# CAT Pruning: Cluster-Aware Token Pruning for Text-to-Image Diffusion Models

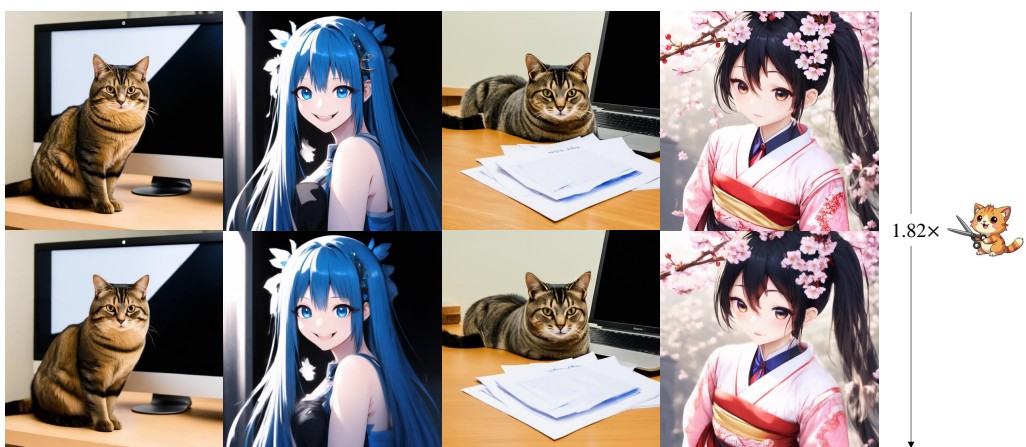

1.82×

Figure 1: **CAT Pruning in Stable Diffusion v3.** The top row depicts the standard denoising process of Stable Diffusion v3 over 28 inference steps, representing the baseline configuration. The bottom row demonstrates the generative performance of CAT Pruning, which achieves similar generative quality while reducing computation cost by 2× and end-to-end inference time by 1.82×.

## ABSTRACT

Diffusion models have revolutionized generative tasks, especially in the domain of text-to-image synthesis; however, their iterative denoising process demands substantial computational resources. In this paper, we present a novel acceleration strategy that integrates token-level pruning with caching techniques to tackle this computational challenge. By employing noise relative magnitude, we identify significant token changes across denoising iterations. Additionally, we enhance token selection by incorporating spatial clustering and ensuring distributional balance. Our experiments demonstrate reveal a 50%-60% reduction in computational costs while preserving the performance of the model, thereby markedly increasing the efficiency of diffusion models.

## 1 INTRODUCTION

Recent advancements in diffusion models (Ho et al., 2020; Dhariwal & Nichol, 2021; Song & Ermon, 2019) have revolutionized generative tasks, especially in the realm of text-to-image synthesis (Karras et al., 2022). Models such as Stable Diffusion 3 (Esser et al., 2024), and Pixart (Chen et al., 2023; 2024) have demonstrated their capability to produce diverse and high-quality images based on user inputs. Despite these successes, the iterative process required for denoising within these models often leads to lengthy and resource-intensive inference periods.

Recent works (Ma et al., 2024b;a) leverage temporal consistency in diffusion models, focusing on the reuse of intermediate features across multiple timesteps. These methods cache features at predetermined timesteps or within specific blocks, thereby reducing computational overhead by reusing

these cached features in subsequent timesteps instead of recomputing them. This approach has proven effective in decreasing the overall computational cost while maintaining generative quality.

While most cache-and-reuse methods focus on bypassing certain `blocks`, thereby reducing the overall number of CUDA kernel launches to save computation, few explore optimizations at the intra-kernel level. Specifically, little attention has been given to reducing the latency within each individual kernel execution.

As mentioned in (Song et al., 2021; Song & Ermon, 2019), Diffusion involves solving a reverse-time SDE using a a time-dependent model. Intuitively, not all patches in an single image require the same precision when it comes to solving the SDE. To further enhance sampling efficiency, we propose a novel acceleration strategy that combines token-level pruning with cache mechanisms. We update a subset of tokens at each iteration, taking into account relative noise magnitude, spatial clustering, and distributional balance.

Our contributions can be listed as follows:

- We observe that token pruning involves ranking token importance while ensuring consistent selection across timesteps and spatial dimensions.

- We propose a simple method that accelerates diffusion models by doing pruning at token level according to relative noise magnitude, selection frequencies, and cluster awareness.

- Our experimental results, evaluated on various standard datasets and pretrained diffusion models, demonstrate that it produces comparable results with 50 % MACs reduction at step 28 and 60 % MACs reduction at step 50 relative to the full size models.

## 2 RELATED WORK

Diffusion models have emerged as powerful generative frameworks in computer vision. However, these models are compute-intensive, often constrained by the high computational cost. This computational bottleneck has led to a surge of research focused on accelerating diffusion models. Here, we highlight three major categories of approaches: parallelization, reduction of sampling steps, and model pruning.

**Parallelization Methods** Despite traditional techniques like tensor parallelism, recent works have introduced novel parallelization strategies specifically tailored to the characteristics of diffusion models. DistriFusion (Li et al., 2024), for instance, hides the communication overhead within the computation via asynchronous communication and introduces displaced patch parallelism, while PipeFusion (Wang et al., 2024c) introduces displaced patch parallelism for Inference of Diffusion Transformer Models (DiT (Peebles & Xie, 2022)) and ParaDiGMS (Shih et al., 2023) rum sampling steps in parallel through iterative refinement.

**Reducing Sampling Steps** One of the core challenges with diffusion models is the large number of sampling steps required to produce high-quality outputs, which directly translates to longer inference times. Recent advancements such as DPM Solver (Lu et al., 2022) and Consistency Models (Song et al., 2023; Song & Dhariwal, 2023) aim to address this bottleneck by developing fast solvers for diffusion ODEs and directly mapping noise to data respectively.

**Leveraging Feature Redundancy** Recognizing the iterative nature of diffusion models and the minimal changes in feature representations across consecutive steps, a growing body of research has focused on developing cache-and-reuse mechanisms to reduce inference time. DeepCache (Ma et al., 2024b) reuses the high-level features of the U-Net (Ronneberger et al., 2015). Block Cache (Wimbauer et al., 2023) performs caching at a per-block level and adjusts the cached values using a lightweight 'scale-shift' mechanism. TGATE (Liu et al., 2024; Zhang et al., 2024) caches the output of the cross-attention module once it converges. FORA (Selvaraju et al., 2024) reuses the outputs from the attention and MLP layers to accelerate DiT inference.

## 3 CAT PRUNING: CLUSTER-AWARE TOKEN PRUNING

Inspired by previous work that accelerates diffusion processes through the exploitation of feature redundancy, we propose cluster-aware token pruning for text-to-image diffusion models, which could synergize with existing methods that implement caching and reuse at the block and module levels.

Applying token-level pruning requires addressing three key challenges. First, we need effective criteria to assess which tokens are critical to the diffusion process. Second, cached features must remain consistent across timesteps to avoid staleness and ensure reliable results. Finally, Token selection should be cluster aware, which means considering spatial structure, to prevent loss of details.

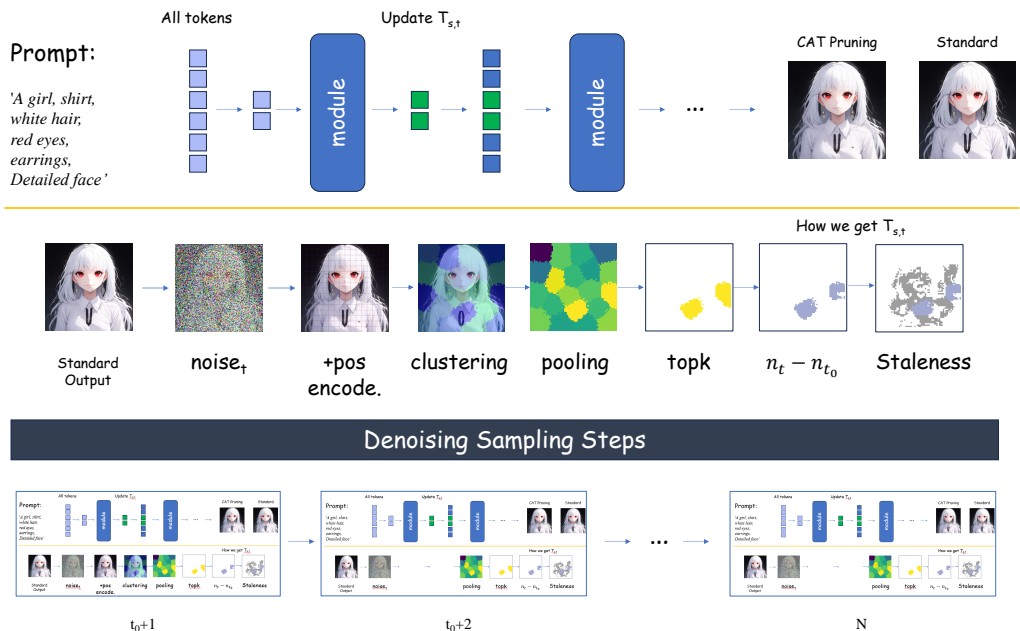

Figure 2: **Method Overview.** At each iteration, tokens are dynamically selected using a combination of the clustering results, noise magnitude, and token staleness. Each part is elaborated in Sec 3.2, Sec 3.3, and Sec 3.4. It is worth noting that we perform clustering only once at step $t_0 + 1$ to avoid computational overhead.

## 3.1 TOKEN PRUNING VIA MASKING

| Notation | Description |
|:---:|:---|
| $h$ | Hidden states |
| $T_{s,t}$ | Tokens selected at the iteration t |
| $T_{u,t}$ | Tokens unselected at iteration t |
| $n_t$ | Noise predicted at iteration $t$ |
| $t_0$ | The step before token pruning starts |
| $f_t$ | A function which maps token to its frequency at step t |
| $N$ | Total denoising steps |
| $\alpha$ | Percentage of tokens being unpruned |

Table 1: Notations used in the paper.

We describe our Algorithm using the notations from Table 1:

**Relative Noise Magnitude** We utilize the variation in noise across timesteps to select tokens. Specifically, we introduce the concept of *Relative Noise Magnitude*, defined as the difference between the current predicted noise and the noise at step $t_0$, which is defined as $n_t - n_{t_0}$ and quantifies the relative change in noise.

Algorithm 1 is an example of how our method applies to the attention mechanism, though it can also be extended to other modules. We use attention here as an illustrative case, and Algorithm 2 describes how we get $T_s$ at each iteration.

---

**Algorithm 1** Attention Forward Pass in CAT Pruning

---

1: $Q, K, V \leftarrow \text{Update}(T_s)$
2: Compute attention:

$$\text{Attention}(Q, K, V) \leftarrow \text{softmax}\left(\frac{QK^\top}{\sqrt{d_k}}\right) V$$

3: **for** $i \in T_{s,t}$ **do**
4:      $h_t[i] \leftarrow MLP(Attention(Q, K, V))$          ▷ Update hidden states of selected tokens
5: **end for**
6: **for** $i \in T_{u,t}$ **do**
7:      $h_t[i] \leftarrow h_{t-1}[i]$          ▷ Reuse hidden states for unselected tokens
8: **end for**

---

### 3.2 CORRELATION BETWEEN PREDICTED NOISE AND HISTORICAL NOISE

Previous work has demonstrated that the changes in **features** across consecutive denoising steps are minimal. This observation motivates our decision to update only a subset of token features at each step, thereby reducing computations.

Furthermore, **PFDiff** (Wang et al., 2024a) has pointed out a notably high similarity in **model outputs** for the existing ODE solvers in diffusion probabilistic models (DPMs), especially when the time step size $\Delta t$ is not extremely large. Building on these two phenomena, we selectively update features for tokens that exhibit substantial changes in their output values, while skipping the feature update and reusing the predicted noise from the previous iteration for the remaining tokens. This reduces computational overhead while maintaining accuracy.

We further observe that the relative magnitude of the noise predicted by the model is correlated with the relative magnitude of historical noise. Specifically, $n_t - n_{t_0}$ is proportional to $n_{t-1} - n_{t_0}$. We demonstrate this by plotting the **L$_2$** norm of relative noise magnitude derived from different prompts and steps in Figure 3.

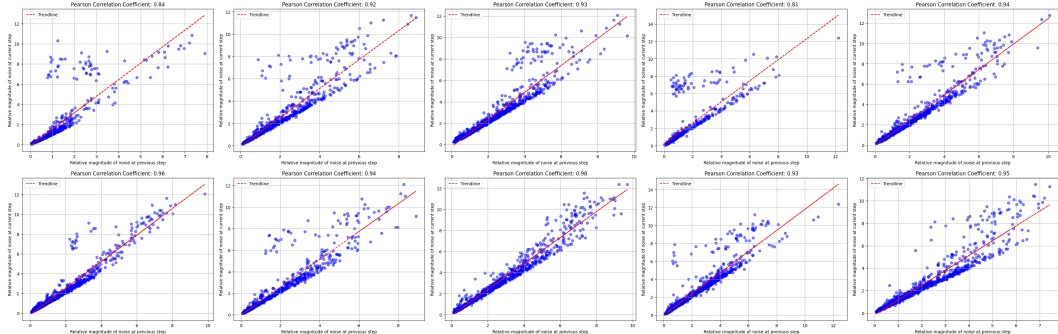

Figure 3: Scatter plot showing the norm of the relative noise at the current step versus the norm of the relative noise at the previous step. We calculate and visualize the Pearson correlation coefficient between these two values.

**Proposition 1.** Selecting tokens with larger relative noise in the current step increases the likelihood that these tokens will exhibit a larger relative noise in subsequent steps.

Given that $t$ is the subsequent step of $t_0$, we provide a proof at timestep $t$ (the simpliest case as for time-step) to substantiate this claim in the appendix.

### 3.3 BALANCING NOISE-BASED TOKEN SELECTION WITH DISTRIBUTIONAL CONSIDERATIONS

In previous iterations, we selected tokens for update based on their relative noise magnitude. While this method is effective in identifying significant changes, it narrows the selected tokens to a specific subset practically.

Inspired by the similarity between the denoising process and SGD (Bottou, 2010), we propose to track the staleness of tokens based on the frequency of each token's selection, which is akin to staleness-aware techniques used in asynchronous SGD algorithms (Dean et al., 2012; Zhang et al., 2015; Zheng et al., 2016).

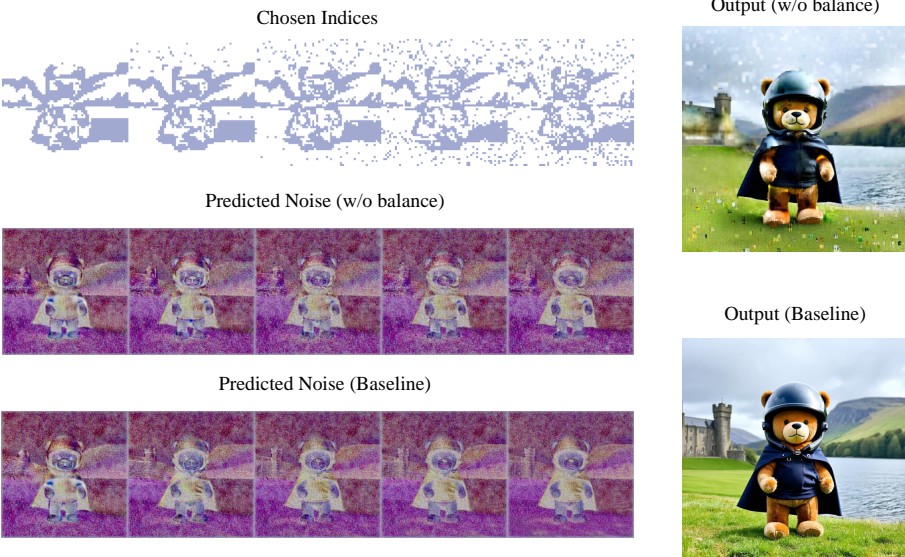

Figure 4: **Visualization of Results Based on Noise Magnitude alone.** Selecting tokens purely by noise magnitude causes the indices to center around the teddy bear's body (as shown in the first row), resulting in noticeable noise artifacts (second row) in the background and a lack of smoothness in the predicted noise.

We visualize the specific indices selected, the predicted noises, and the final generated images when tokens are chosen solely based on the magnitude of change. As shown in Figure 4, repetitively focusing on certain tokens degrades the overall image by introducing inconsistencies and unbounded staleness.

Following the *exploration and exploitation* (Auer et al., 2002; Sutton & Barto, 1998) trade-off commonly used in reinforcement learning (RL) algorithms, we propose a more distributional-balanced (also staleness-aware) selection strategy. By incorporating the trade-off manually, we ensure that while tokens with significant noise changes are given certain priority, there is still a promising exploration of other tokens.

For the *exploration* part, we perform **Frequency Monitoring** track the selection of each token. To be more specific, we employ an exponentially weighted moving average (EWMA) to prioritize recent selections over earlier ones when measuring frequency:

$$f_0 = I_0, \tag{1}$$
$$f_n = a \times f_{n-1} + I_n, \tag{2}$$

where $f_t$ shows the moving average at integer time $t \geq 0$, and $I_t$ is an indicator function that equals 1 when the token is selected at step $t$. The *exploitation* part continues to use $n_t - n_{t_0}$ as a criterion.

As shown in Figure 5, considering the staleness of each token leads to smoother output noise and a final image that closely resembles the one generated by the full-size model.

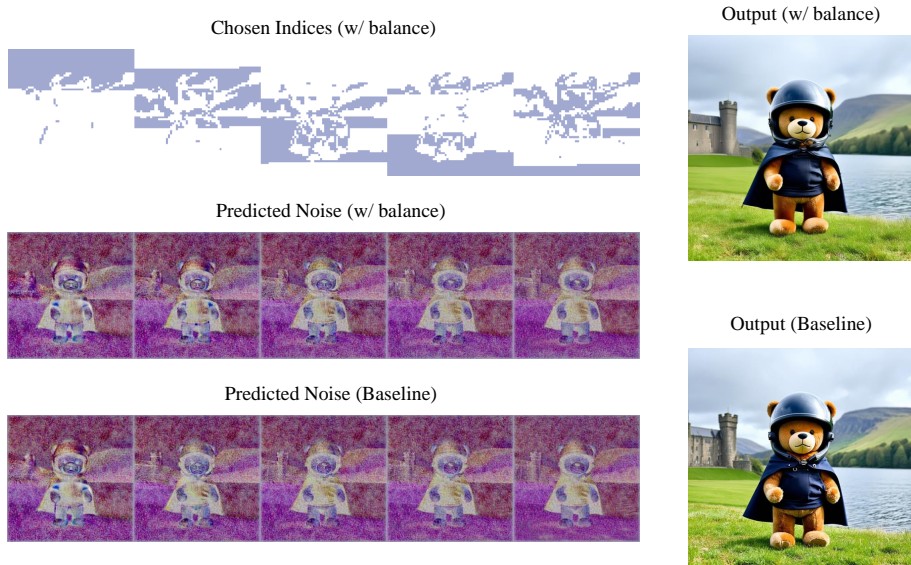

Figure 5: **Visualization of Results Based on Noise Magnitude and Token Staleness.** Incorporating both staleness and noise magnitude in token selection yields a more balanced selection distribution, resulting in improved outputs with notably smoother backgrounds and smoother predicted noises.

### 3.4 CLUSTERING GIVES RISE TO MORE SPATIAL DETAILS

So far, our ultimate goal is to approximate the output image using token pruning. However, it happens that toknes at certain positions tend to change synchronously across iterations.

We observe that tokens with spatial adjacency tend to require similar selection frequencies. To verify this, we perform an ablation study where consecutive rows of the output are selected at each iteration (i.e. step $t_0 + 1$: row 1,2, step $t_0 + 2$: row 3,4). As demonstrated in Figure 6, a simple sequential token selection strategy (column 3) yields strong results, even when masking 70% of the tokens. Furthermore, incorporating clustering information (column 2) enhances detail preservation compared to its non-clustering counterpart, outperforming the naive sequential strategy. For example, in row 1, column 1, there is a lack of windows; in row 1, column 3, the windows appear blurry. In row 2, column 1, there is an inconsistent smile; in row 2, column 3, the heart is missing. However, column 2 does not have these issues, as it maintains spatial consistency and incorporates many details.

Therefore, we maintain that the proposed pruning algorithm should also take the spatial co-relation into account so as to better appoximate the final output. To achieve this requirement, questions arise such as:

1. How should we split the output into several spatial co-related clusters?

2. What value should we grant each spatial cluster?

3. How to perform token selection inside each cluster?

**Enforcing Spatial-awareness while Clustering** Simple clustering is agnostic to spatial relations, which is essential to the performance. There are several approaches for spatial-aware clustering on graphs, including graph cuts (Shi & Malik, 1997) and GNN-based methods (Bianchi et al., 2020). We opt for positional encodings to enforce spatial-awareness due to their simplicity and low computational overhead. Our customized Positional Encoding is formulated as:

$$pos\_enc(i \cdot w + j, :) = \begin{cases} \frac{i}{h}, & \text{if } 1 \leq k \leq \frac{d}{2} \\ \frac{j}{w}, & \text{if } \frac{d}{2} + 1 \leq k \leq d \end{cases} \tag{3}$$

{noise, balance}   {cluster, noise, balance}  {consecutive rows}                Baseline

Figure 6: **Comparative Analysis of Token Selection Strategies.** The first colomn displays images generated by selecting tokens based on noise magnitude and distributional balance. The second colomn incorporates clustering information for enhanced spatial coherence. The third colomn shows results from a naive strategy of sequential token selection. All three strategies have pruned 70% tokens, where $t_0 = 8, N = 28$.

where $i$ and $j$ denote the row and column, respectively, and $d$ is the dimension of the noise magnitude.

After adding this positional encoding, we perform KMeans (MacQueen, 1967) using $L_2$ as the clustering metric. We visualize the clustering (n=20) results of several different prompts in Figure 7.

**Graph Pooling Fosters Inter-cluster Consistency** Meanwhile, we hope the value of each cluster preserves the feature of specific patches as well as its neighbors, and therefore we introduce 1 light-weighted Graph Pooling Layer, which is not trainable.

**Preserves Distributional Balance within Each Cluster** Practically, we notice that it's also beneficial to introduce distributional balance inside each cluster. Thus for each selected cluster(each with $\sim 200$ tokens, we choose tokens according to their noise magnitude as well as their selection frequencies. This part is not included in Algorithm 2 just for simplicity.

**Summary of Algorithm**

Our algorithm proceeds as Algorithm 2:

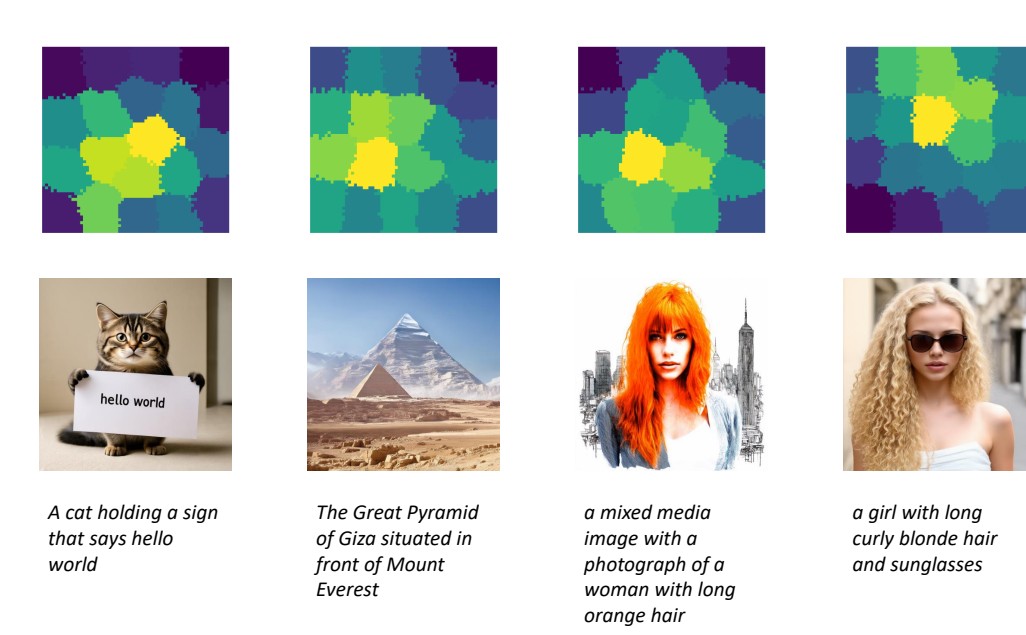

*A cat holding a sign that says hello world*

*The Great Pyramid of Giza situated in front of Mount Everest*

*a mixed media image with a photograph of a woman with long orange hair*

*a girl with long curly blonde hair and sunglasses*

Figure 7: **The clustering results of different prompts.** For each token, clustering is performed based on its relative noise magnitude with positional encoding. We use the K-means algorithm with $L_2$ distance as the clustering metric.

---

**Algorithm 2** Finding Indices for CAT Pruning

1: **Input:** $i, t_0, n_i, n_{t_0}$
2: $indices \leftarrow []$
3: $RN \leftarrow n_i - n_{t_0}$
4: **if** $i == t_0 + 1$ **then**
5:     $clusters \leftarrow KMeans(pos\_enc + n_i - n_{t_0})$                 ▷ Cluster noise
6:     $graph\_scores \leftarrow pool(clusters, n_i - n_{t_0} + pos\_enc)$    ▷ Aggregate cluster scores
7:     $top\_clusters \leftarrow topk(graph\_scores)$
8:     **for** each $c \in top\_clusters$ **do**
9:         $indices \leftarrow indices \cup topk((n_i - n_{t_0})[j], \text{ for } j \in c)$
10:     **end for**
11: **else**
12:     $graph\_scores \leftarrow pool(clusters, pos\_enc + n_i - n_{t_0})$    ▷ Use clusters from $t_0 + 1$
13:     $top\_clusters \leftarrow topk(graph\_scores)$
14:     **for** each $c \in top\_clusters$ **do**
15:         $indices \leftarrow indices \cup topk((n_i - n_{t_0})[j], \text{ for } j \in c)$
16:     **end for**
17:     $indices \leftarrow indices \cup topk(-f_i(j), for j \notin indices)$        ▷ Add stale tokens
18: **end if**
19: **return** $indices$

---

## 4 EXPERIMENTS

### 4.1 SETUPS

**Models** We evaluate our method on several pretrained Diffusion Models: Stable Diffusion v3 and Pixart-$\Sigma$, which feature superior performance of text-to-image synthesis over various metrics.

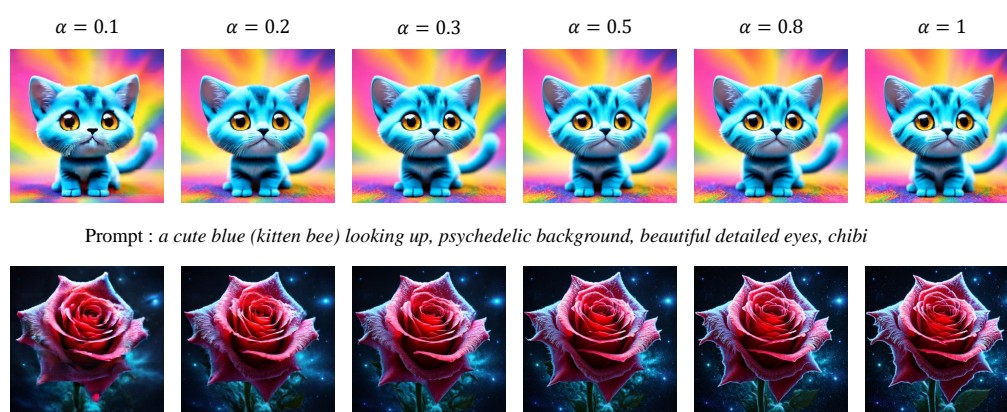

Prompt : *a cute blue (kitten bee) looking up, psychedelic background, beautiful detailed eyes, chibi*

Prompt : *a frozen cosmic rose, the petals glitter with a crystalline shimmer, swirling nebulas, 8k unreal engine photorealism, ethereal lighting, red, nighttime, darkness, surreal art*

Figure 8: **Qualitative Results with different sparsity and different prompts.** In these cases, even $\alpha = 0.2$ gives strong results.

**Datasets** We select datasets that are adopted to evaluate text-to-image tasks, including MS-COCO 2017(Lin et al., 2015) and PartiPrompts (Yu et al., 2022), which contain 5K prompts and 1.6K prompts respectively.

**Implementation Details** We evaluate our methods using 28 and 50 sampling steps, respectively. For both Stable Diffusion 3 and Pixart-$\Sigma$. We employ classifier-free guidance (Ho, 2022) with guidance strengths of 7.0 and 4.5, consistent with their official demo settings. All inferences are performed in float16 precision on a single Nvidia A5000 GPU. Both models generate images at a resolution of $1024 \times 1024$, reflecting real-world scenarios.

**Baselines** We use both the output of the standard diffusion model and AT-EDM (Wang et al., 2024b) as baselines, and the latter is a token pruning technique. For AT-EDM, we implement its algorithm under the same token budget with our algorithm, which is starting token pruning at step 9 and pruning 70 % tokens at each iteration. Specifically, since AT-EDM is actually designed for SD-XL, which utilizes token pruning and similarity-based copy, and in practical 30% token budget is not suitable for similarity-based copy, so we combine the token selection algorithm in AT-EDM and the cache-and-reuse mechanism as a baseline.

### 4.2 MAIN RESULTS

**Analysis of Different Levels of Sparsity** In Figure 8 and Figure 9, we present visualizations of generated images across various prompts and sparsity levels, characterized by the percentage of un-pruned tokens, denoted as $\alpha$. As $\alpha$ increases, the generated content progressively approximates that of the full-sized model. Notably, there is little perceptible difference between $\alpha = 0.3, 0.5, 0.8$, and $\alpha = 1.0$ (the standard diffusion model output). However, at $\alpha = 0.2$, degradation becomes evident, such as the reduced number of windows and a missing eye in in Figure 9 compared to the standard output. Based on these observations, we select $\alpha = 0.3$ as the optimal value for all subsequent evaluations, striking a balance between model performance and computational efficiency.

**Speedups** The results in Tab. 2 demonstrate the performance of our method at 28 sampling steps. For Stable Diffusion 3 on the PartiPrompts dataset, we achieve a significant reduction in total computation, from 168.28T to 90.28T, yielding a $1.82\times$ speedup while maintaining a comparable CLIP Score (Radford et al., 2021; Hessel et al., 2022). Similarly, for Pixart-$\Sigma$, our method delivers a $1.73\times$ speedup with negligible impact on CLIP Score.

We further evaluate our method under the $N = 50$ setting: we could achieve about $2\times$ speedup while maintaining the overall performance and getting better CLIP Score compared to AT-EDM(Wang et al., 2024b).

| $\alpha = 0.1$ | $\alpha = 0.2$ | $\alpha = 0.3$ | $\alpha = 0.5$ | $\alpha = 0.8$ | $\alpha = 1$ |
|---|---|---|---|---|---|

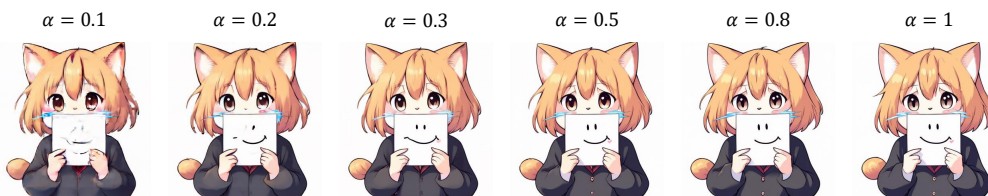

Prompt : *beautiful anime artwork, a cute anime catgirl that looks depressed holding a piece of paper with a smile drawn on it over her mouth, she is about to cry*

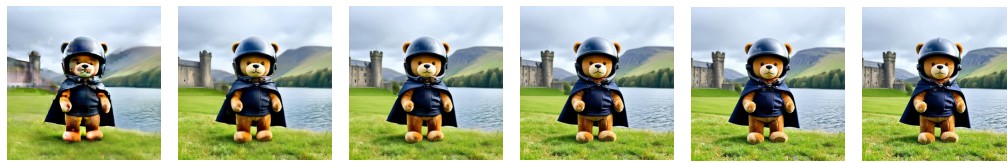

Prompt : *A teddy bear wearing a motorcycle helmet and cape is standing in front of Loch Awe with Kilchurn Castle behind him*

Figure 9: **Qualitative Results with different sparsity and different prompts.** We find $\alpha = 0.3$ a sweet spot for the tradeoff between computation efficiency as well as the image quality.

| Method | PartiPrompts | | | | COCO2017 | | | |
|---|---|---|---|---|---|---|---|---|
| | MACs ↓ | Throughput ↑ | Speed ↑ | CLIP Score ↑ | MACs ↓ | Throughput ↑ | Speed ↑ | CLIP Score ↑ |
| SD3 - 28 steps | 168.28T | 0.119 | 1.00× | 32.33 | 168.28T | 0.113 | 1.00 × | 32.47 |
| Ours - 28 steps | 90.28T | 0.217 | 1.82× | 32.03 | 90.28T | 0.212 | 1.87 × | 32.21 |
| AT-EDM - 28 steps | 93.48T | 0.166 | 1.40× | 31.07 | 93.48T | 0.170 | 1.51× | 30.59 |
| Pixart-$\Sigma$ - 28 steps | 120.68T | 0.151 | 1.00× | 31.12 | 120.68T | 0.143 | 1.00× | 31.36 |
| Ours - 28 steps | 60.08 T | 0.262 | 1.73 × | 31.06 | 60.08 T | 0.258 | 1.80× | 30.02 |
| AT-EDM - 28 steps | 62.08T | 0.238 | 1.57 × | 24.30 | 62.08T | 0.244 | 1.71× | 14.66 |

Table 2: Comparison of different methods on PartiPrompts and COCO2017 datasets. All methods here adopt 28 sampling steps.

| Method | PartiPrompts | | | | COCO2017 | | | |
|---|---|---|---|---|---|---|---|---|
| | MACs ↓ | Throughput ↑ | Speed ↑ | CLIP Score ↑ | MACs ↓ | Throughput ↑ | Speed ↑ | CLIP Score ↑ |
| SD3 - 50 steps | 300.50 T | 0.062 | 1.00 × | 32.92 | 300.50 T | 0.062 | 1.00× | 32.20 |
| Ours - 50 steps | 136.70 T | 0.134 | 2.15 × | 32.72 | 136.70 T | 0.130 | 2.08 × | 32.18 |
| AT-EDM - 50 steps | 143.42T | 0.107 | 1.72 × | 28.48 | 143.42T | 0.102 | 1.64× | 28.20 |
| Pixart-$\Sigma$ - 50 steps | 215.40T | 0.079 | 1.00× | 31.41 | 215.40T | 0.078 | 1.00 × | 31.20 |
| Ours - 50 steps | 88.24 T | 0.166 | 2.09 × | 31.36 | 88.24 T | 0.160 | 2.04 × | 30.62 |
| AT-EDM - 50 steps | 92.44T | 0.148 | 1.87 × | 17.08 | 92.44T | 0.147 | 1.88 × | 11.00 |

Table 3: Comparison of different methods on PartiPrompts and COCO2017 datasets 50 Steps. All methods here adopt 50 sampling steps.

## 5 CONCLUSION

In this paper, we introduce a novel acceleration strategy for diffusion models that combines token-level pruning with cache mechanisms. By selectively updating a subset of tokens at each iteration, we significantly reduce computational overhead while preserving model performance.

Our experiments demonstrated that the proposed method effectively maintains generative quality, achieving up 50% reduction in MACs at 28-denosing-step and 60 % at 50-denosing-step. We evaluated our approach on standard datasets and pretrained diffusion models, showing that it produces results comparable to the original models.

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

# A APPENDIX

## A.1 PROOF OF PROPOSITON 1.

Let $T_{s,t-1}$ and $T_{u,t-1}$ denote the selected and unselected token sets, respectively. At step $t-1$, we assume:

$$\mathrm{n}[i], \quad i \in T_{s,t-1} > \mathrm{n}[i], \quad i \in T_{u,t-1}$$

For the hidden states $h$ at step $t$:

$$h_t[T_{u,t}] = h_{t-1}[T_{u,t}],$$
$$h_t[T_{s,t}] = \texttt{Update}(T_{s,t})$$

Thus, $h$ for the unselected tokens remains unchanged, while the selected tokens are changed based on their current activations using a model specific function `Update`.

From this, we have:

$$\mathrm{MSE}(h_t, h_{t-1})[i], \quad i \in T_{\mathrm{s,t}} > \mathrm{MSE}(h_t, h_{t-1})[i], \quad i \in T_{\mathrm{u,t}}$$

Since the predicted noise is a function of the hidden states, the magnitude of the predicted noise relative to noise at step $t_0$ is directly tied to the change in hidden states.

As a result, at each step, we select tokens based on their relative noise magnitude.

