# OpenReview forum: "CAT Pruning: Cluster-Aware Token Pruning For Text-to-Image Diffusion Models"
_ICLR.cc/2025/Conference — Submitted to ICLR 2025_

### Official Review · Reviewer_upgQ · 2024-11-03

**Soundness:** 2
**Presentation:** 2
**Contribution:** 2
**Rating:** 5
**Confidence:** 3

**Summary:**

This paper proposes a new method for accelerating text-to-image diffusion models by selectively updating a subset of tokens during the denoising process. The authors introduce Cluster-Aware Token Pruning (CAT Pruning), which leverages the relative noise magnitude of tokens, their selection frequencies, and spatial clustering to achieve significant computational savings while maintaining the quality of generated images. They demonstrate that CAT Pruning can achieve up to a 50% reduction in computational costs with minimal impact on image quality, making diffusion models more efficient for generating high-resolution images. The paper also provides extensive experimental results on popular datasets and pretrained diffusion models, comparing CAT Pruning to existing methods and highlighting its superior performance.

Key contributions of the paper include:
Proposing A Token Importance Ranking Procedure: The paper establishes a method for ranking token importance that considers not only noise magnitude but also selection frequencies across timesteps, ensuring consistent token selection.
A Cluster-Aware Pruning Method: The authors propose a unique pruning method that integrates spatial clustering, leveraging positional encoding to maintain spatial coherence and detail preservation in generated images. This approach improves the quality of outputs compared to simple sequential token selection strategies.
Making the Case for Distributional Balance: The paper emphasizes the importance of distributional balance within clusters. This balance is achieved by considering both noise magnitude and selection frequencies when selecting tokens within each cluster. This contributes to a more nuanced pruning process that avoids over-emphasizing certain features at the expense of others.

**Strengths:**

Originality: The paper presents a novel approach called CAT Pruning, which combines token-level pruning with caching techniques to accelerate text-to-image diffusion models. While previous works have explored caching and reuse mechanisms to reduce inference time, CAT Pruning focuses on optimizing at the intra-kernel level by reducing latency within individual kernel executions. The authors introduce the concept of “Relative Noise Magnitude” to identify significant token changes across denoising iterations. This concept is defined as the difference between the current predicted noise and the noise at step t0, which is defined as nt−nt0 and quantifies the relative change in noise They also incorporate spatial clustering and ensure distributional balance to enhance token selection, further improving efficiency and preserving model performance.

Quality: The paper demonstrates a high level of quality through a reasonable amount of experimentation and analysis. The authors evaluate CAT Pruning on standard datasets like MS-COCO 2017 and PartiPrompts, using established pretrained diffusion models such as Stable Diffusion v3 and Pixart-Σ. They compare their method against relevant baselines, including the standard diffusion model output and AT-EDM, another token pruning technique. The results show significant reductions in computation costs (up to 50% reduction in MACs at 28 denoising steps and 60% at 50 denoising steps—although the authors should actually define the acronym MACs) while maintaining comparable or even superior CLIP scores. The authors provide visualizations of generated images at different sparsity levels, demonstrating the effectiveness of CAT Pruning in preserving image quality even with significant pruning. They also offer insights into the correlation between predicted noise and historical noise, justifying their token selection strategy.

Clarity: The paper is relatively well-written and structured, presenting the proposed method in a clear and concise manner. The authors provide a reasonable overview of the problem and related work, highlighting the limitations of existing approaches. They clearly define some key notation in Table 1 and describe the algorithm using illustrative examples and figures. The experimental setup is detailed, allowing for reproducibility and a clear understanding of the evaluation process. The results are presented in tables and visualized through figures, facilitating interpretation and analysis.

Significance: The paper addresses a crucial challenge in the field of text-to-image synthesis: the high computational cost of diffusion models. By significantly accelerating inference time without compromising image quality, CAT Pruning has the potential to make these powerful generative models more accessible for various applications. This work contributes to the growing body of research on optimizing diffusion models and could inspire further advancements in efficiency and scalability. The authors’ insights into token-level pruning and the exploitation of feature redundancy could benefit other generative tasks beyond text-to-image synthesis.

**Weaknesses:**

The paper has limited theoretical justification: The paper primarily relies on empirical observations and intuitions to justify the effectiveness of CAT Pruning. While the authors present Proposition 1 and provide a simplified proof in the appendix, a more rigorous theoretical analysis could strengthen the paper's contribution.  A deeper theoretical understanding of the relationship between relative noise magnitude, token staleness, and spatial clustering could lead to more informed design choices and potentially improved performance. For example, exploring the convergence properties of the algorithm or deriving bounds on the error introduced by pruning would provide valuable insights.

There is a lack of comparison with other pruning techniques, and other techniques in general: The paper compares CAT Pruning only with AT-EDM, another token pruning technique. However, a comprehensive comparison with a wider range of pruning methods for diffusion models, such as those leveraging model distillation, quantization, or low-rank factorization, would provide a more complete picture of the proposed method's strengths and limitations. This would allow for a more informed assessment of the relative performance and efficiency of CAT Pruning compared to other state-of-the-art techniques. In particular the paper seems to ignore the highly influential VQVAE and VQGAN based methods for leveraging compressed latent representations of data. These methods are extremely popular ways for reducing the computational load of diffusion models. See Gu et al., (2022) for example (among many other examples), i.e.

Gu, S., Chen, D., Bao, J., Wen, F., Zhang, B., Chen, D., Yuan, L. and Guo, B., 2022. Vector quantized diffusion model for text-to-image synthesis. In Proceedings of the IEEE/CVF conference on computer vision and pattern recognition (pp. 10696-10706).

"CAT Pruning" focuses on optimizing token processing within a U-Net architecture. However, VQ-Diffusion demonstrates that utilizing a VQVAE to learn a compressed, discrete latent representation can lead to significant computational savings. By shifting the diffusion process to this lower-dimensional latent space, VQ-Diffusion achieves notable speed improvements.
The "CAT Pruning" paper does not acknowledge or compare its approach to this architectural shift towards latent space diffusion using VQVAEs, which constitutes a notable weakness. Similarly, the paper on "High-Resolution Image Synthesis with Latent Diffusion Models"  of

Rombach, R., Blattmann, A., Lorenz, D., Esser, P. and Ommer, B., 2022. High-resolution image synthesis with latent diffusion models. In Proceedings of the IEEE/CVF conference on computer vision and pattern recognition (pp. 10684-10695).

discusses the popularity and advantages of VQGANs, particularly their ability to learn compressed latent representations that can be used for high-resolution image synthesis. VQGANs, in conjunction with autoregressive models, have emerged as powerful tools for high-resolution image synthesis. These models operate on a compressed, discretized latent space learned by a VQGAN, potentially offering significant computational advantages over pixel-based approaches. The "High-Resolution Image Synthesis" paper uses VQ-regularization as one method for training the autoencoder that produces the latent space for their LDMs. They discuss how their model "can be interpreted as a VQGAN but with the quantization layer absorbed by the decoder."

Since the proposed CAT Pruning technique emphasizes reducing computational costs as a primary goal, comparing this CAT token pruning method with VQVAE and VQGAN style approaches in a much more direct manner would be very helpful. These methods inherently operate in a compressed latent space, and understanding the relationships between these kinds of methods and the proposed method would be very helpful, comparing empirically would be strongly desired and it could demonstrate the relative efficiency of CAT Pruning much more clearly.

In summary, with respect to the issue of comparing the work here to these popular methods, the paper would be strengthened by:
1.	Explicitly at least acknowledging the popularity of VQVAE and VQGAN-based approaches for diffusion model acceleration.
2.	Discussing the potential trade-offs between their token pruning within the U-Net architecture and the use of VQGANs and other approaches for latent space diffusion.
3.	Ideally, conducting more direct experiments to compare the performance, efficiency, and image quality of CAT Pruning against a VQGAN-based diffusion model.

**Questions:**

Questions:

Regarding VQVAEs and Latent Space Diffusion: The "Vector Quantized Diffusion Model for Text-to-Image Synthesis" paper presents VQ-Diffusion, a method that uses a VQVAE to perform diffusion in a compressed latent space. This approach achieves significant speed improvements.

* Given the potential efficiency gains of VQVAEs for diffusion, could the authors explain their rationale for focusing on token pruning within the U-Net architecture and not comparing with those methods, conceptually or emprically?
* What are the perceived advantages and disadvantages of each approach?
* Considering the importance of VQGANs in this domain, why weren't they included as a baseline for comparison in "CAT Pruning"?
A key focus of "CAT Pruning" is computational efficiency.
* Could the authors provide a more direct comparison of the efficiency gains of CAT Pruning against VQGAN-based diffusion models and LDMs?
This would involve metrics like inference time, memory usage, and the number of floating-point operations.

Suggestions:

While Proposition 1 is presented, a more comprehensive theoretical foundation for CAT Pruning would strengthen the paper. Pfdiff provides a very detailed analytical explanation for why their approach takes the decisions that it does. Brining the work here closer to that level of analytical analysis would be helpful. Some ideas for how that might be achieved include:
Deriving bounds on the error introduced by pruning.
Exploring the relationship between noise magnitude, staleness, and clustering in more depth.

As a point of comparison "Deep Cache" emphasizes that the high-level features generated during the reverse diffusion process exhibit significant temporal consistency. This observation forms the basis for their caching mechanism, which avoids redundant computations. This work could benefit from explicitly acknowledging and discussing the role of temporal redundancy in the effectiveness of their method. Perhaps one could explain how the relative stability of certain tokens across timesteps (as captured by the "staleness" metric) might relate to the temporal consistency observed in "Deep Cache." "Deep Cache" includes comparisons with various baselines, including pruning and distillation methods. "CAT Pruning" could benefit from a similarly more comprehensive evaluation. For example the work here could directly compare "CAT Pruning" with "Deep Cache" to assess their relative performance and efficiency.

AT-EDM introduces a Denoising-Steps-Aware Pruning (DSAP) schedule that adjusts pruning ratios across different denoising timesteps. This schedule prunes fewer tokens in early steps when attention maps are less informative and more aggressively in later steps when redundancy is higher. CAT Pruning also acknowledges the varying importance of denoising steps and implements a prune-less schedule in early steps. Some kind of discussion and comparison of these different approaches and their motivations could be helpful.

While the paper mentions the use of existing caching techniques, a more in-depth discussion of token recovery strategies, particularly in the context of subsequent convolutional layers, would strengthen the paper. Exploring alternative methods, such as the similarity-based copy technique proposed in AT-EDM, could further improve the effectiveness and generalizability of CAT Pruning.

---

### Official Review · Reviewer_nXeT · 2024-11-03

**Soundness:** 3
**Presentation:** 3
**Contribution:** 2
**Rating:** 5
**Confidence:** 3

**Summary:**

The paper (CAT Pruning) introduces a technique to increase the inference efficiency of diffusion models. In particular, the authors combine caching mechanisms with token-wise pruning which reduces the computational overhead during inference. The paper is overall well written and provides good quantitative results for latest text-to-image models (such as SD-3), however lacks some performance benchmarking against some baselines and a more holistic evaluation of generation beyond the CLIP-score.

**Strengths:**

Below I state the strengths and weaknesses of the paper:

- The paper is well-written and easy to follow — with a strong motivation. Moreover improving the efficiency during inference, is a practical problem which still has some open problems for the newer t2i models based on the transformer architecture.
- The analysis on the token selection strategy based on relative noise magnitude (combined with spatial clustering and balancing) is comprehensive.

**Weaknesses:**

Weaknesses:

- [Minor] The authors are suggested to provide a brief overview of the diffusion model architecture (e.g., SD-3 like architecture) to which their method is applicable.  This will improve readability by quite an extent.
- [Major] The paper does not compare with some of the caching only baselines (e.g., TGATE which they cite). This would be important to understand the effectiveness of combining caching with pruning techniques. Although not directly comparable, I’d also suggest the authors to provide the distillation based baselines (less number of steps in the inference) in the paper, to provide a full picture of the effectiveness of the caching + pruning family of methods.
- [Major] The authors provide the CLIP-score for generation quality — but it’d be more effective to show a more holistic evaluation of the method in terms of generation quality. For example, the authors can test on compositionality, long caption generation etc for their method in terms of generation.
- [Minor]: The authors are suggested to provide more qualitative results comparing the generation across different methods improving efficiency.

**Questions:**

See Weakness.

Overall, the paper introduces a technically solid method, but lacks some important comparisons on evaluations. I am happy to revisit my score during the rebuttal, if the authors respond to the Weaknesses adequately.

---

### Official Review · Reviewer_9Tq1 · 2024-11-04

**Soundness:** 2
**Presentation:** 1
**Contribution:** 2
**Rating:** 3
**Confidence:** 4

**Summary:**

The paper presents CAT Pruning, a method to increase the computational efficiency of diffusion model sampling. CAT pruning targets the attention calculations of the DM and aims to eliminate irrelevant tokens. The method combines clustering results, noise magnitude and staleness of tokens to identify unimportant tokens. The results indicate no perceiptable loss of output information when pruning up to 70% of tokens which results in a speedup of up to 60%.

**Strengths:**

- **Important topic.** Computational efficenciy is one of the main limitations of Diffusion Models. Works addressing these issues are of high value to the field.
- **Good performance** CAT Pruning appears to preseve the original performance of the model well, while yielding a decent speedup.

**Weaknesses:**

The paper has two overarching weaknesses that become apparent in a multitude of smaller issues.

**Presentation**
Key aspects of the paper are presented badly, making it hard for readers to grasp the contributions made by CAT-Pruning
- Underlying fundamentals of the method are not sufficiently explained. For example nowhere in the Abstract, Introduction or Conclusion to the authors elaborate that they prune tokens in the attention layers of the DM. In fact when asking multiple computer vision researchers what they assumed the method to be about, all of them assumed that tokens where pruned in the embeddings of the text prompt.
- The authors use MACs as a key performance metric throughout the paper without ever explaining it
- Similarly it is never explained between which samples the CLIP Score is calculated during evaluation
- a third of Page 7 is just empty, but Page 8 is almost exclusively Figures
- Tables 2 and 3 do not contain any bold numbers
- The diffusion notations are somewhat disconnected from common conventions. For example, noise (estimates) are usually reffered to as $\epsilon$, during generation diffusion steps $t$ should **decrease** since this is the reverse diffusion process. The authors also claim that Diffusion involves solving a reverse-time SDE, which is indeed one valid mathmatical foundation for diffusion. However, it specifically does not encompas the models actually used in the remainder of the paper, with SD3 being a rectified flow model that is specifically incompatibel with stochhastic algorithms.


**Evaluation**
Further, there are problems with the validation of the poposed method.
- The main claim is that the output image after pruning is perceptually similar to the image generated without pruning. However, that specific aspect is never empirically evaluated. The obvious choice here would have been to simply report LPIPS distance between the pruned und unpruned image.
- Similarly the tradeoff between compute reduction and (un)pruning percentage $\alpha$ is demonstrated with 4 examples but not ablated over empirically
- In the same vain many of the design choices of the CAT algorithm are showcased with 1 or 2 qualitiative examples with limited to no empirical ablations. A more structured analysis on the importance/downstream influence of the different components in the selection algorithm would have been important
- The empriical results in Tab. 2 and 3 do not contain any confidence intervals or standard deviations
- There is no qualitative comparison with competing methods
- The authors only compare against one other baseline, although other methods exist, including ∆-DiT [1], Faster Diffusion [2], or TGATE [3] or basic methods like KV caching
- The most prominent choices for speeding up DM inference are, of course, distillation methods or consistency models. Consequently, it would be important to consider if CAT-Pruning still offers advantages when applied to distilled or consistency models.

[1] Pengtao Chen, et al.  ∆-DIT: A training-free acceleration method tailored for diffusion transformers. arXiv:2406.01125
[2] Senmao Li et al. Faster diffusion: Rethinking the role of unet encoder in diffusion models.  arXiv:2312.09608, 2023.
[3 Wentian Zhang, et al. Cross-attention makes inference cumbersome in text-to-image diffusion models. arXiv:2404.02747, 2024.

**Other**
 - Method appears to be limited to DiT architecture. At least no other architecture is considered

**Questions:**

- **Q1.** Is CAT-Pruning restricted to DiTs or does it also apply to other architectures like UNet DMs with attention?
- **Q2.** You write that "cached features must remain consistent across timesteps" L 110 and that your methods combines token-level pruning with ache mechanisms L 64. How exactly is the cache optimization realized with CAT-Pruning? At no point in the paper do you mention which part of the method actually is responsible for the cache optimiztion. Is that achieved by the Frequency monitoring over timesteps?

---

### Official Review · Reviewer_3Lg7 · 2024-11-05

**Soundness:** 2
**Presentation:** 2
**Contribution:** 2
**Rating:** 3
**Confidence:** 3

**Summary:**

This paper introduces CAT Pruning (Cluster-Aware Token Pruning), an acceleration technique for text-to-image diffusion models that aims to reduce computational cost by selectively updating tokens based on relative noise magnitude, spatial clustering, and balanced selection frequencies. By combining token pruning with caching, CAT Pruning demonstrates up to a 2x speedup and a 50-60% reduction in computation on two models (Stable Diffusion 3 and Pixart-Σ), two denoising steps (28 and 50), and two datasets (PartiPrompts and COCO2017), while maintaining CLIP score.

**Strengths:**

1. The proposed method achieves notable speedup (up to 2x) while preserving CLIP Score.
2. The proposed method can be applied to pre-trained models without additional training costs, making it a lightweight option for improving inference efficiency across different tasks.

**Weaknesses:**

1. The paper lacks a thorough discussion of prior token pruning work, making it difficult to assess the proposed method’s novelty and improvements over existing methods. A more detailed overview of token pruning techniques, their limitations, and how the proposed method addresses these would clarify its contributions.
2. The study lacks comparisons with a diverse set of token pruning baselines and established training-free methods (e.g., caching), which limits a comprehensive view of the proposed method’s effectiveness relative to existing techniques.
3. The qualitative differences shown in Figure 6 between clustering and non-clustering approaches are subtle, and there is no rigorous ablation study to quantitatively assess clustering’s impact on model performance. A thorough ablation study is needed to substantiate the claimed benefits of clustering.
4. The paper relies solely on CLIP Score to assess image quality, which measures text-image alignment but not visual fidelity. Including metrics such as FID would provide a more complete evaluation of image quality and support claims of fidelity preservation.

**Questions:**

1. Could you clarify where the difference in inference speed arises between CAT Pruning and existing token pruning methods?
2. What is t0 used in the experiments?
3. Could you explain where the differences in speed and CLIP score arise between CAT Pruning and the AT-EDM baseline?

---

### Meta-Review · Area_Chair_oxuW · 2024-12-18

**Metareview:**

This paper studies the inference efficiency issue for the diffusion model. The main idea is to identify the important tokens in the diffusion steps and only updates those tokens during the inference time. This observes a remarkable speedup gain.

The reviewers praise this paper for:
1. notable speedup and important topic
2. easy to follow writing.

The reviewers argue that:
1. lacking enough comparison / baselines.
2. lacking enough discussion b/w links to the previous token pruning works.
3. presentation issue (the background / context of this paper is not fully explained / introduced, which makes it hard to read in the beginning)
4. CLIP based benchmark is not thorough to guarantee the quality of the generated results.

Given the lack of supports from the reviewers, I would suggest reject.

**Additional Comments On Reviewer Discussion:**

There is no rebuttal provided by the authors.

---

### Decision · Program_Chairs · 2025-01-22

Reject